# Wasted *Ganoderma tsugae* Derived Chitosans for Smear Layer Removal in Endodontic Treatment

**DOI:** 10.3390/polym11111795

**Published:** 2019-11-01

**Authors:** Sheng-Tung Huang, Nai-Chia Teng, Hsin-Hui Wang, Sung-Chih Hsieh, Jen-Chang Yang

**Affiliations:** 1Department of Chemical Engineering and Biotechnology, Institute of Biochemical and Biomedical Engineering, National Taipei University of Technology, Taipei 106-08, Taiwan; ws75624@ntut.edu.tw; 2Research Center of Biomedical Device, Taipei Medical University, Taipei 110-52, Taiwan; 3School of Dentistry, College of Oral Medicine, Taipei Medical University, Taipei 110-31, Taiwan; tengnaichia@hotmail.com; 4Department of Dentistry, Taipei Municipal Wan-Fang Hospital, Taipei 116-96, Taiwan; 97240@w.tmu.edu.tw; 5Graduate Institute of Nanomedicine and Medical Engineering, College of Biomedical Engineering, Taipei Medical University, Taipei 110-31, Taiwan; 6International Ph.D. Program in Biomedical Engineering, College of Biomedical Engineering, Taipei Medical University, Taipei 110-31, Taiwan; 7Research Center of Digital Oral Science and Technology, Taipei Medical University, Taipei, 110-52, Taiwan

**Keywords:** EDTA, polysaccharide, chelating effect, smear layer removal

## Abstract

The objective of this study is to investigate the synergistic effects of acid etching and metal-ion chelation in dental smear layer removal using wasted *Ganoderma tsugae* derived chitosans. The wasted *Ganoderma tsugae* fruiting body was used to prepare both acid-soluble fungal chitosan (FCS) and alkali-soluble polysaccharide (ASP). To explore the effective irrigant concentration for smear layer removal, a chelating effect on ferrous ions was conducted. Specimens of various concentrations of EDTA, citric acid, and polysaccharide solutions were reacted with FerroZine™ then the absorbance was examined at 562 nm by a UV-visible spectrophotometer to calculate their metal chelating capability. Twenty extracted premolars were instrumented and individually soaked in the solutions of 15 wt% EDTA, 10 wt% citric acid, 0.04 wt% ASP, 0.04 wt% FCS, and normal saline were randomly divided into five groups (N=4). Next, each tooth was cleaved longitudinally and examined by scanning electron microscopy (SEM) to assay the effectiveness of smear layer removal. The chelating capability for EDTA, FCS, and ASP showed no significant difference over the concentration of 0.04 wt% (*p* > 0.05). The SEM results showed that 0.04 wt% FCS solution was effective in smear layer removal along the canal wall. These results indicated that *Ganoderma tsuage* derived FCS in acid solutions could be a potential alternative as a root canal irrigant solution due to its synergistic effect.

## 1. Introduction

The goal of endodontic treatment is to clean the root canal system and eliminate the microorganisms as well as necrotic pulp tissue remnants to prevent reinfection of pulpal/periradicular pathosis [1]. Root canal disinfection is usually achieved by mechanical instrumentation and chemical irrigation solution due to the complexity of root anatomy [2]. Smear layer is a thin layer of pulverized mixture comprising of dentin, pulp, and bacterial remnants found spread on root canal walls after instrumentation [3]. Comprehensive literature reviews for smear layer and its clinical implications and relevance to endodontics were reported by Torabinejad et al. [4]. The removal of the smear layer was believed to be beneficial in improving disinfection and good adaptation of dental resin composites to the canal walls. Gu et al. [5] offered an extensive review about irrigant agitation techniques and devices. Many smear layer removal methods were proposed such as chemical [6], ultrasonic [7], and laser [8] techniques, but none of them are totally effective throughout the length of all canals [9].

In operative dentistry, combining ethylenediaminetetraacetic acid (EDTA) with sodium hypochlorite (NaOCl) is the most widely used method in smear layer removal [10]. The chelation of 17 wt% of EDTA (pH 8) with calcium ions is believed to play a major role in promoting decalcification of dentine [11]. However, EDTA is a substituted diamine with cytotoxicity and weak genotoxicity. Oral exposure to EDTA causing adverse reproductive and developmental effects in animals was reported [12]. The 10 wt% citric acid is an alternative EDTA-free irrigant in smear layer removal without any chelating capability reported by Leonardo et al. [13]. It is effective in smear layer removal based on the high solubility of dental hard tissue in acidic environments. To design an irrigant system with high solubility of hydroxyapatite and metal-chelating capability, we proposed the usage of an acid-soluble cleating agent. Among natural polymers, chitosan and its derivatives are known as renewable resources with high chelating capacity and versatile chemical modification [14].

*Ganoderma tsugae*, known as *Ling-Zhi* in Chinese, is an important fungus with several biological activities in traditional Chinese medicine [15]. The major cell wall content of fungi is chitin (1-4 β-poly(*N*-acetylglucosamine)). Unlike the chitosan produced from crustacean chitin with non-reproducible qualities due to seasonal variation and crustacean species, fungal mycelia are relatively consistent in composition and considered to be free from allergenic animal antigens [16,17]. Currently, the process of alkali-soluble polysaccharide (ASP) prepared from the mild alkaline digesting of *G. tsugae* fruiting body residue is established in our laboratory. Taking advantage of flexibility in preparing acid-soluble fungal chitosan (FCS) and ASP from the same *G. tsugae* fruiting bodies, the decoupling of metal-ion chelation and dissolution mechanisms in smear layer removal becomes possible. The aim of this study was to investigate the possible mechanisms in dental smear layer removal for future optimal irrigant solution formulation. 

## 2. Materials and Methods

*G. tsugae* residue was obtained from a local mushroom farm in Chiayi County, Taiwan. The stepwise preparation flow diagram of FCS and alkaline deacetylation for ASP are shown in Figure 1a. The reagents for deactylation of *G. tsugae* residues were sodium hydroxide (NaOH, Macron Fine Chemicals, Selangor, Malaysia) and acetic acid (CH_3_COOH, Wako Pure Chemical Co., Osaka, Japan). The detailed preparation procedures of FCS were reported elsewhere [18].

The acid-soluble fungal chitosan (FCS) was dissolved in 2% lactic acid at 40 °C for 4 h. Intrinsic viscosities were measured using an automated iVisc Capillary viscometer (Lauda-Königshofen, Germany). The measurement was carried out at 25 °C. The capillary K is 0.009869 mm^2^/s^2^. Finally, the viscosity average molecular weights of chitosan were calculated using the Mark–Houwink equation.
[η] = KMv^α^(1)
where K of 16.80 × 10^−3^ mL/g and α of 0.81 were the viscosity parameters used in this calculation [19]. The resulting FCS had a degree of deacetylation (DD) value of 83.7% and molecular weight (MW) of 104 kDa; while ASP revealed DD values of 93.7% and a MW of 43 kDa.

### 2.1. Chelating Ability

The reagent grade of ferrous chloride (FeCl_2_·H_2_O, J.T. Baker, Japan), citric acid (C_6_H_8_O_7_, Sigma-Aldrich, Tokyo, Japan), ethylenediaminetetraacetic acid, disodium salt, dihydrate (Na_2_EDTA·2H_2_O) (C_10_H_16_N_2_O_8_, J.T. Baker, Chu-Bei City, Taiwan), and FerroZine^TM^ iron reagent (3-(2-pyridyl)-5,6-bis(phenyl sulfonic acid)-1,2,4-triazine, Acros Organics, Gleel, Belgium) for the chelating effect measurement were used as received without purification. Chelating ability was determined according to the method of Riemer et al. [20]. Each chitosan sample (0.1–10 mg/mL) in 2 g/L acetic acid solution (0.2 mL) was mixed with 0.02 mL of 2 mM ferrous chloride. The reaction was initiated by the addition of 0.02 mL of 5 mM FerroZine^TM^, then shaken vigorously and left at room temperature for 10 min. The absorbance of the mixture was measured by a UV–Visible spectrophotometer (HITACHI, U-2001, Tokyo, Japan) at 562 nm against a blank according to the FerroZine^TM^-based colorimetric assay. A lower absorbance indicates a higher chelating power. Citric acid and EDTA were used for comparison. Chelating effect was calculated using the following equation:(2)$$Chelatingeffect\;(\%)=[1-\frac{OD_{562nm,sample}}{OD_{562nm,blank}}]\times 100$$

### 2.2. Smear Layer Removal

Root canals of access cavities in 20 extracted premolars were prepared by the crown-down technique with rotary nickel-titanium files (K3^®^; SybronEndo Corporation, Orange, USA) and irrigated with 5 wt% sodium hypochlorite by a 25-gauge blunt-end needle of syringe. The anatomic diameter was determined by introducing successively larger K-files to the working length until resistance was felt upon removal of the file. To investigate the smear layer removal effect of the final irrigation, 20 canals were randomly divided into five groups (N=4) and irrigated with 15 wt% EDTA, 10 wt% citric acids, 0.04 wt% of FSC and ASP solutions, and control (normal saline) for 1 min. All irrigated canals received a final rinse with distilled water for 5 min to halt any chemical activity. The crowns were removed at the cementum–enamel junction. The roots were then split longitudinally with a chisel and a hammer. Finally, specimens were gold-coated and examined by scanning electron microscopy (SEM) (Hitachi S2400, Tokyo, Japan) under an accelerating voltage of 15–20 kV.

### 2.3. Statistical Analysis

Student’s t-test was used to evaluate the statistical significance of the measurement data. The results were considered statistically different at *p* < 0.05.

## 3. Results

The chelating effect can be quantitatively determined by measuring the rate of color reduction. The absorbance decreased linearly with increasing concentrations of coexisting chelating agents. Figure 2 is the concentration dependence on ferrous ion chelating ability for various irrigant solutions determined colorimetrically. The percentages of chelating activities increased as solution concentration increased. Citric acid showed little chelating ability for ferrous ions within the concentration less than 0.05 wt%. At the concentration of 0.01 wt%, the chelating ability of EDTA, ASP, and FCS were 100%, 52%, and 28%, respectively. The sequence of chelation capability for various solutions was EDTA > ASP > FCS > citric acid when the concentration was less than 0.03 wt%. However, EDTA, ASP, and FCS revealed similar chelation capabilities when the concentration was beyond 0.03 wt%. Based on these findings, the 0.04 wt% of acidic type of FCS solution and basic type of ASP were chosen to compare the smear layer cleaning efficacy to 10 wt% citric acid and 15 wt% EDTA to correlate the effects of the solubility property and chelating capability.

Figure 3 represented the SEM photographs of the root canal surface treated by various irrigation solutions. After treating with 15 wt% EDTA or 10 wt% citric acid, the SEM of middle root canal third (×350) showing most of the smear layer was removed, and some open tubule orifices covered with visible debris of the smear layer on the intertubular and peritubular dentine (Figure 3a,b). Removal of the entire smear layer from dentin surface but with certain severe dentin surface erosion was observed in Figure 3c for the 0.04 wt% FCS solution group. However, the smear layer in the middle third of the root canal was observed after irrigation with 0.04 wt% ASP or normal saline (Figure 3d,e). The efficacy of smear layer removal for 0.04 wt% FCS, 15 wt% EDTA, and 10 wt% citric acid was better than that of 0.04 wt% ASP and normal saline for each concentration indicated in Figure 3.

## 4. Discussion

Effective smear-layer removal plays a pivotal role in the successful outcome of endodontic therapy [21]. Due to the similar smear layer cleaning efficacy between 10 wt% citric acid and 15 wt% EDTA, we are interested in knowing what causes the hydroxyapatite (HAp) breakdown shown in SEM.

### 4.1. Acid Etching Effect

It is important to have a basic understanding about how irrigants interact with HAp. Hydroxyapatite is a crystal with the characteristic of ionic bonds. During the dissolution process, HAp loses its ordered arrangement of the solid mineral phase and becomes free in aqueous solution. Based on the mechanism of the solubility product principle, the water molecules were responsible for disrupting the crystal lattice bonds [22]. In acid dissolution, H^+^ ions react with PO_4_^3−^ and OH^−^ ions, which results in reducing their concentration, and then more solid HAp dissolves until the solubility product is re-established, reaching the dissolution equilibrium of HAp [23].

Another mechanism based on a chemical reaction claimed that H^+^ ions, not water molecules, were attributed as the primary crystal lattice-disrupting agent [24]. According to the chemical equation of Equation 1, H^+^ ions in solution approach the apatite crystal and react with PO_4_^3−^ and OH^−^ ions at the surface of the solid. Conversion to HPO_4_^2−^ and H_2_O causes a disruption of lattice bonds and release of the ions into solution.

*Ca*_10_(*PO*_4_)_6_(*OH*)_2_ + 8*H*^+^ → 10*Ca*^2+^ + 6*HPO*_4_^2−^ + 2*H*_2_*O*(3)

H^+^-induced conformation changes in the collagen matrix might result in contractile stress (ca. 0.2–0.4 MPa) development that was sufficient to cause a collapse of the demineralized dentin matrix [25].

### 4.2. Chelation Effect

Chelation effect usually involves the formation of multiple coordinate bonds between a multiple bonded ligand (electron pair donor) and a metal ion (electron pair acceptor). EDTA, a hexaprotic weak acid (H_6_Y^2+^) with four carboxylic acids and two ammoniums [26], was first proposed by Nygaard-Ostby to facilitate root canal preparation by chelation [27]. When EDTA is mixed with water, it has an acidic pH, but cannot be used for irrigation as such due to poor solubility. In addition, the action mechanism of EDTA is co-existent with protonation and chelation depending on the pH value as described below [28]:(a) **Protonation:** EDTA-H^−3^ + H^+^ → EDTA-H_2_^−2^ (pH < 7)(4)

(b) **Chelation:** EDTA-H^−3^ + Ca^+2^ → EDTA-Ca^−2^ + H^+^ (pH > 7)(5)

At a low pH, the protonation of EDTA will reduce the available electro pairs and retard the dissociation of HAp as well as demineralization. On the contrary, the EDTA binding of calcium ions will tend to increase the dissociation of HAp and its availability for chelation at a high or neutral pH. The pH of EDTA has to be adjusted up to a minimum of pH 7 or above to enhance the water solubility but decreases the protonation effect. EDTA chelates calcium ions and forms soluble calcium chelates. When the disodium salt of EDTA is added to this equilibrium, calcium ions are removed from the solution. This leads to the dissolution of further ions from dentin so that the solubility product remains constant.

A continuous rinse with 5 mL of 17% EDTA, as a final rinse for 3 min, efficiently removes the smear layer from root canal walls [29]. But some studies showed that even 1% EDTA solution has a good chelating power [30]. Von der Fehr and Nygaard-Ostby found that EDTA decalcified dentine to a depth of 20 to 30 μm in 5 min [11]. According to Saito et al. greater smear layer removal was found in the 1-min EDTA irrigation group than the 30-sec or 15-sec groups [31]. No clear-cut guideline about the optimal time period for aqueous EDTA solution irrigation was addressed [21]. Prolonged exposure of root dentine to EDTA might have the potential risk of root dentine weakening [32]. Steward et al. found that the combination of EDTA with urea peroxide is very effective in cleaning root canals due to the instrumentation capacity enhancement [33].

### 4.3. Acid Etching Plus Chelation Effect

Native chitosan is insoluble in water, but its solubility is significantly enhanced when the pH is below its isoelectric point (PI = 6.3) [34]. The chelation of chitosan with metal ions is attributed to the complex formation through the amine groups functioning as ligands [35]. Ion-chelating ability of chitosan is strongly affected by the degree of deacetylation, while fully acetylated chitosan showing little chelating activity [36].

Silva et al. investigated the time-dependent effects of chitosan/acetic acid solution on dentin [37]. Removal of the entire smear layer from the dentin surface was observed with 0.1 wt% chitosan for 3 min, while the severe dentin surface erosion was evidenced by the tubular diameter increasing and extensive destruction of the intertubular dentin in 0.37 wt% chitosan for 5 min. To quantify the efficacy of smear layer removal using chitosan compared with different chelating agents, the concentration of calcium ions in the solutions after irrigation were examined by atomic absorption spectrophotometry with flame [38]. The calcium ion concentration difference between the specimens treated with 15% EDTA and 0.2% chitosan was not significant.

In this study, the group of 0.04 wt% FCS in acetic acid solution revealed complete removal of the entire smear layer from the dentin surface and even with some erosion on the dentin surface showing its high efficiency over 15 wt% EDTA or 10 wt% citric acid. Even with similar chelating ability to FCS, but acid-nonsoluble, 0.04 wt% ASP showed poor smear layer removal outcome. It is evident that both the acid etching and chelation effect are effective for smear layer removal. Furthermore, combining 0.04 wt% FCS in acetic acid solution produces a synergistic effect over the sum of their individual effects.

Like *G. tsugae*, extracts from many herbal/medicinal plants also show antioxidant activity and chelating ability [39]. With the understanding of the interplay between decalcification by metal-ion chelation and acid etching, the measurement of chelating ability offers a good starting point to develop a formulation with more biocompatibility but that is less aggressive to the periapical tissues.

## 5. Conclusions

The cleaning efficacy of 0.04 wt% FSC was higher than that of 15 wt% EDTA and 10 wt% citric acid. The chitosan dissolved in acid solution offered a synergistic effect of acid etching and chelation on smear layer removal. It suggests that cell wall derivatives from fungal biomasses might provide an alternate irrigant suitable for clinical use.

## Figures and Tables

**Figure 1 polymers-11-01795-f001:**
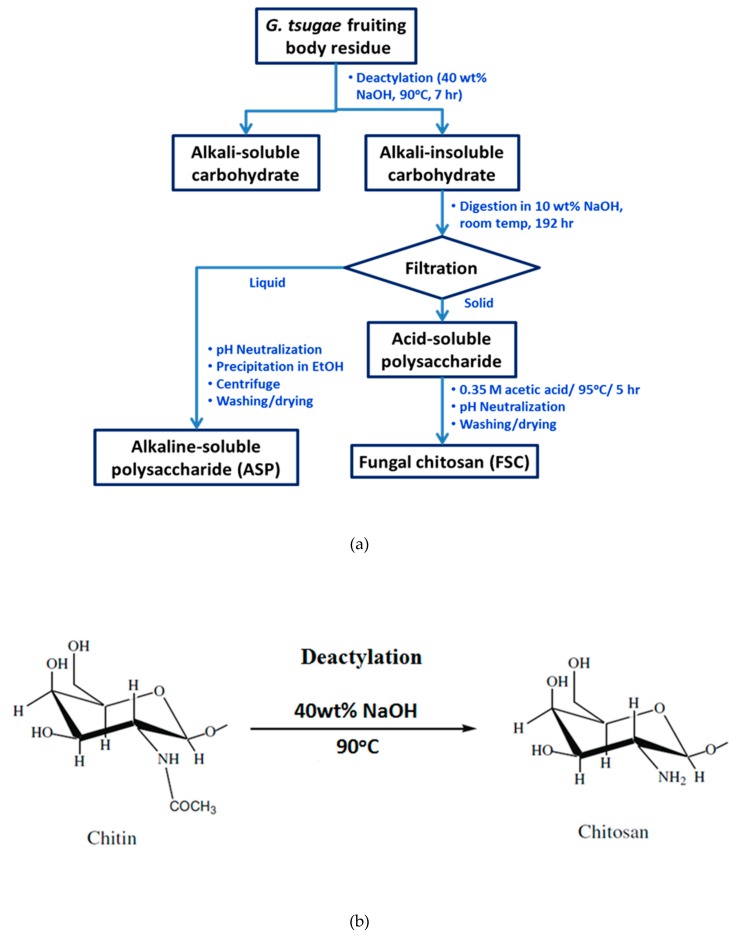
Flow diagram for alkali-soluble polysaccharide (ASP) and fungal chitosan (FSC) preparation: (**a**) flow diagram of ASP and FSC preparation; (**b**) deactylation of chitin for preparation of chitosan.

**Figure 2 polymers-11-01795-f002:**
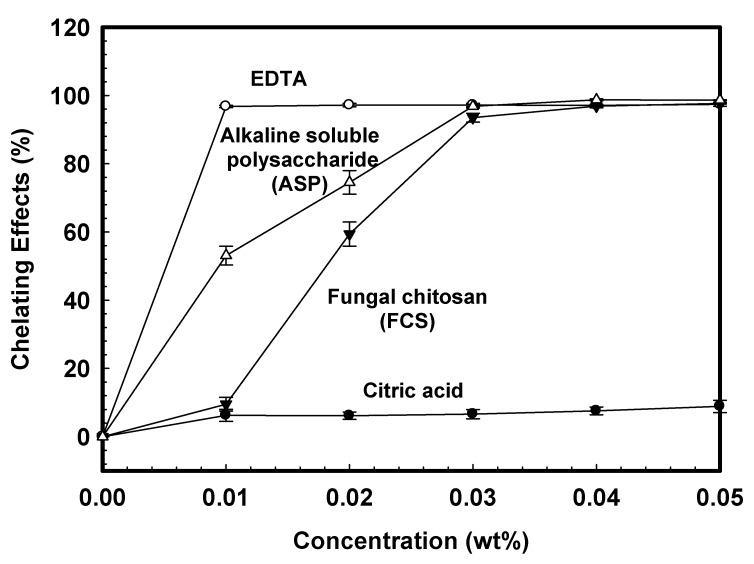
The concentration dependence of chelating effects for various irrigation solutions.

**Figure 3 polymers-11-01795-f003:**
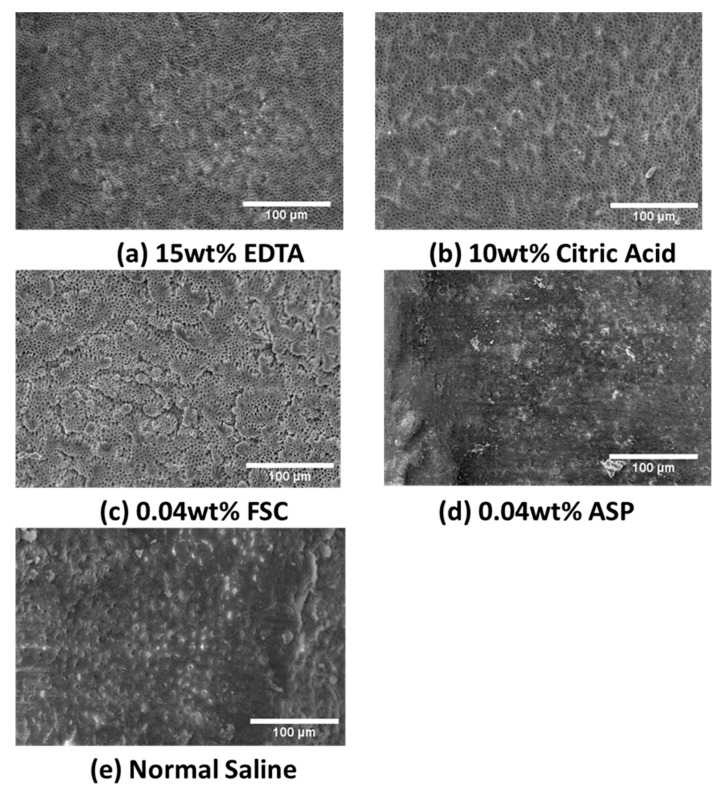
The representative scanning electron microscopy (SEM) photographs (350×) of the prepared surface treated by various irrigation solutions: (**a**) 15 wt% ethylenediaminetetraacetic acid (EDTA); (**b**) 10 wt% citric acid; (**c**) 0.04 wt% fungal chitosan (FCS); (**d**) 0.04 wt% alkali-soluble polysaccharide (ASP); (**e**) normal saline.

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
