# Peer review of "Wasted Ganoderma tsugae Derived Chitosans for Smear Layer Removal in Endodontic Treatment"

_polymers, 2019, doi:10.3390/polym11111795_

Round 1
Reviewer 1 Report
Dear editor,
This manuscript is well-written and well organised. It is an interesting study.
The introduction provide sufficient background and include all relevant references (maybe recent review or book chapter should be included to improved the quality of the revised mnuscript. The methods are sufficiently described in the manuscript. The research design and results are appropriate and clearly presented and discussed in the manuscript. For all these reasons, this present manuscript could be accepted for publication after minor revisions (english revision and reference list revision).
Regards,
Reviewer 2 Report
In this revised version, the authors have addressed all my questions and concerns. I recommend this work for publication.
Author Response
Thanks for reviewer's kind approval.
This manuscript is a resubmission of an earlier submission. The following is a list of the peer review reports and author responses from that submission.
Round 1
Reviewer 1 Report
The major disadvantage of this study is the poor discussion. The results are adequate but the discussion and analysis are poor and nor comparable to already published literature. Also, authors should further emphasize on the novelty of their work. The level of English language is too poor and needs intense revision by a native English. Also, the reference list can be even more updated (more recent relative works).
Reviewer 2 Report
Dear editor,
The manuscript submitted by Sheng-Tung Huang et al. entitled: “Wasted Ganoderma tsugae Derived Chitosans for Smear Layer Removal in Endodontic Treatment” aims to study the extraction of chitosan derivatives from fungus. All things considered it is an interesting study, which can be accepted for publication in Polymers, after major revision. Please to see my comments below to improve the revised manuscript.
Comments:
- Author should reformulate the title of this publication in the revised manuscript. This title is too confused for reader.
- Authors must give more keywords in the revised manuscript (i.e. Chitosan, Fungus).
- In introduction part, authors have made statements on chitosan and not backed it up with references concerning the production and the applications of fungal chitosan largely described in litterature.
- In result and discussion part, authors must give more comparison with literatures and with chitosan from crustaceans.
- In conclusion part, the aspects of novelty and the putative future valorization and applications of fungal chitosan should be more underlined by authors by comparison with chitosan from crustaceans.
General comment:
In the revised manuscript, the authors need to pay more attention to grammatical construction of sentences and spelling of sentences! Authors must give another figure 1.b. to represent chitosan in the revised manuscript. In fact, chitosan is this study is not a full deacetylated form. Why authors did not use commercial crustacean chitosan for comparison?
Reviewer 3 Report
This research work investigate the synergistic effects of acid etching and metal-ion chelation in dental smear layer removal using wasted Ganoderma tsugae derived chitosans. The reported results are relevant to the scientific community, and the work contains important research contributions. The article can be published after minor revision.
Page 2 Line 73
It is stated that: "The aim of this study was to investigate the possible mechanisms in dental smear layer removal for future optimal irrigant solution formulation." Although the paper demonstrate the efficiency of the 0.04 wt% FSC on the removal of smear layer, no mechanisms explaining the synergistic effect are discussed. The authors should explain the efficacy of smear layer removal in terms of the mechanisms.
Page 1 Line 43
Please remove the extra parenthesis in Reference [1)]
Page 7 Line 155
Please rephrase the sentence “According to the chemical equation of Equation 1,…”